# Temperature Incubation Influences Gonadal Gene Expression during Leopard Gecko Development

**DOI:** 10.3390/ani12223186

**Published:** 2022-11-17

**Authors:** Maria Michela Pallotta, Chiara Fogliano, Rosa Carotenuto

**Affiliations:** 1Institute for Biomedical Technologies, National Research Council, 56124 Pisa, Italy; 2Department of Biology, University of Naples Federico II, Via Cinthia 26, 80126 Naples, Italy

**Keywords:** lizard, sex determination, incubation temperature, gonadal differentiation

## Abstract

**Simple Summary:**

Environmental sex determination is a modality of sex determination related to external factors and that has implicated determinants such as climatic conditions, which act on the embryo after fertilization and deposition of the egg. For reptiles, the temperature is the main element for sex determination; this factor affects laid eggs in different ways. Details remain to be elucidated concerning the temporal gene expression and the functions of their protein products. Therefore, the aim of the present work was to determine the genetic determinants differentially represented during the embryonic development of a model species already known in temperature-dependent sex determination, the leopard gecko *Eublepharis macularius*. Following this investigation, new data were acquired on genes expressed in the sexual differentiation of *E. macularius*. In addition, new genes potentially involved in the mechanisms of tissue and metabolic sexual differentiation of the embryo of this species have been identified. This study could bring new useful information in order to correctly interpret the regulatory pathway underlying the determination of sex in vertebrates.

**Abstract:**

During development, sexual differentiation results in physiological, anatomical and metabolic differences that implicate not only the gonads but also other body structures. Sex in Leopard geckos is determined by egg incubation temperature. Based on the premise that the developmental decision of gender does not depend on a single gene, we performed an analysis on *E. macularius* to gain insights into the genes that may be involved in gonads’ sexual differentiation during the thermosensitive period. All the genes were identified as differentially expressed at stage 30 during the labile phase of sex differentiation. In this way, the expression of genes known to be involved in gonadal sexual differentiation, such as *WNT4*, *SOX9*, *DMRT1*, *Erα*, *Erβ*, *GnRH*, *P450 aromatase*, *PRL*
*and PRL-R*, was investigated. Other genes putatively involved in sex differentiation were sought by differential display. Our findings indicate that embryo exposure to a sex-determining temperature induces differential expression of several genes that are involved not only in gonadal differentiation, but also in several biological pathways (*ALDOC*, *FREM1*, *BBIP1*, *CA5A*, *NADH5*, *L1 non-LTR retrotransposons*, *PKM*). Our data perfectly fit within the new studies conducted in developmental biology, which indicate that in the developing embryo, in addition to gonadal differentiation, sex-specific tissue and metabolic polarization take place in all organisms.

## 1. Introduction

In many organisms, sex is determined by the presence of heteromorphic chromosomes and by factors encoded by them, which establish balances between specific regulatory patterns. In mammals, for example, the SRY factor linked to the sex chromosome Y is decisive for testicular differentiation [1]: in these cases, we speak of genotyping sex determination (GSD). Several cases of GSD exist: one of the best known is that of *D. melanogaster* [2], in which sex determination depends on the ratio of the number of X sex chromosomes to the number of autosomes A. Sexual development in mammals, on the other hand, is a more complex process and is independent of the ratio of the number of sex/autosomal chromosomes present in the individual’s genome but originates from the presence/absence of the Y chromosome [3], from which derives a transcript, Sex-determining region Y protein (*SRY*), which, by interacting with factors such as SRY-box containing gene 9 (*Sox9*), *Sf1* and *Dmrt1* (dsx and mab-3-related transcription factor 1) [4] controls the differentiation of the bipotential gonad in the male direction in embryos with XY chromosomes. There is another possibility of sex determination, which is dependence on environmental factors (environmental sex determination—ESD). The latter is related to factors external to the organism’s genome and sees implicated determinants, such as climatic conditions, which act on the embryo after fertilization and deposition of the egg [5]. Reptiles, unlike some birds and mammals, exhibit sex determination that is both dependent on sex chromosomes and dependent on external factors [6]. The modes of GSD include at least three different conditions of heterogametes. When we talk about environmental determinants, we refer, for reptiles, to temperature as the main one; this factor affects laid eggs in different ways. The prerequisite for the persistence of ESD in reptiles seems to be related to the thermally heterogeneous environment, in which the natural and different incubation temperatures of the different microenvironments in which eggs are laid ensure the determination and development of both gonads and the physiological characterization of the male and female sexes. Granting that temperature can act as a motive for the activation of specific pathways that induce sex differentiation [7], it is possible to assume the existence of a period during egg incubation when the embryo is temperature sensitive (TSP) [8]. Although several genetic factors are important for sex determination and are regulated by temperature, details remain to be elucidated in regard to temporal gene expression and the functions of their protein products [9].

Reptiles are among the organisms that tolerate temperature variations worst [10,11]. To date, however, information on the gene and molecular network that would guide gonadogenesis in these species remains limited, both regarding the different components and regarding their respective functions. Even in mammals, the lack of data on all genes expressed during the early stages of gonadal development has limited the ability to delineate the complete pathway of genes that would regulate the early stages of ovary development [12,13]. The aim of the present work was mainly to individuate which are the genetic determinants differentially represented during the embryonic development of a model species, the leopard gecko (*Eublepharis macularius*), a well-known model in the study of the mechanisms of embryonic development and temperature-dependent sex determination [14,15,16]. The possible variation in expression of genes involved in gonad differentiation and in other pathways was then evaluated. All the genes were identified as differentially expressed at stage 30 during the labile phase of sex differentiation [17]. For our aims, we used differential display (DDRT-PCR), and the results obtained were validated by means of real-time PCR. Since the use of model species has proved useful for identifying new genetic factors and understanding their mechanisms of action, the study of reptiles could bring new useful information in order to correctly interpret the regulatory pathway underlying the determination of the sex in vertebrates.

## 2. Materials and Methods

### 2.1. Animals 

Two females and one male specimen of *E. macularius* were housed in a terrarium at the Department of Biology of Università di Napoli “Federico II”, according to the institution’s Animal Welfare Office guidelines and policies and to international rules and to the recommendations of the Guide for the Care and Use of Laboratory Animals of the American National Institutes of Health and of the Italian Health Ministry. The experimental protocol was approved by the institutional Animal Experiments Ethics Committee (Centro Servizi Veterinari) (permit number: 2014/0017970). Fertilization occurred naturally. Every month (May, June and July), after fertilization, each female deposed two eggs. Each experiment was technically replicated three times. Each pair of eggs (6 pairs, 12 eggs in total) were collected and immediately placed in two precision incubators (±0.1 °C) set to a constant temperature of 26 °C (FPT, six eggs) or 32.5 °C (MPT, six eggs), for 7 days, roughly corresponding to stage 30 [17]. Temperature and moisture were monitored daily using HOBO temperature loggers (Onset Computer Corporation, Pocassett, MA, USA).

### 2.2. RNAs

Each embryo (*N* = 12; 6 from MPT and 6 from FPT derived from two females and deposed in different months) were eviscerated, and the area strictly adjacent to the not fully formed gonads was taken. Total RNA was extracted from each single embryo according to the TRI-Reagent protocol (Sigma Aldrich, Saint Louis, MO, USA). The concentration and purity of RNA samples were determined by UV absorbance spectrophotometry; RNA integrity was checked by 2.0% agarose gel electrophoresis. RNA extracted from the 12 embryos at stage 30, the stage in which the undifferentiated gonad is sensible to the temperature [17], was subdivided between qRT-PCR (*N* = 6; 3 from MPT and 3 from FPT) and DDRT-PCR (*N* = 6; 3 from MPT and 3 from FPT). First-strand cDNA, used for all amplification reactions, was synthesized from singularly extracted RNA from each MPT and FPT embryo, then utilized to obtain 1 μg of total RNA using Super Script III Reverse Transcriptase (Invitrogen, Waltham, MA, USA) and used for the two screening protocols [16]. 

### 2.3. Expression Analysis of Genes Involved in Gonadal Sex Differentiation 

Differential expression analysis of nine genes involved in gonadal sex differentiation, estrogen receptor α (*Erα*), estrogen receptor β (*Erβ*), gonadotropin-releasing hormone (*GnRH*), P450 aromatase, prolactin (*PRL*), prolactin receptor (*PRL-R*), Wnt family member 4 (*WNT4*), sex determining region Y-Box 9 (*SOX9*) and doublesex and Mab-3 related transcription factor 1 (*DMRT1*) was carried out by quantitative RT (qRT) PCR using primers designed on vertebrate sequences found in GenBank. Sequences of interest were aligned by using a Multiple Sequence Alignment free software (http://www.genome.jp/tools-bin/clustalw, accessed on 10 November 2022) and primers designed on the sequence regions with the highest degree of identity by means of Primer 3Plus software (http://www.bioinformatics.nl/cgibin/primer3plus/primer3plus.cgi, accessed on 10 November 2022) (Table 1) [16]. 

### 2.4. DDRT-PCR 

Differential display allows one to compare and identify changes in gene expression at the mRNA level between two or more cell populations. Briefly, RNA was reverse transcribed using anchored oligo-dT primers designed to specifically bind to the 5′ ends of the poly-A tails. Successively, cDNAs were amplified by using the anchored oligo-dT primers in combination with a series of arbitrary 5‘ primers and amplification products, then separated and visualized by electrophoresis. For our purpose, RT-PCR and PCR were performed using the RNA spectra kit and fluorescent mRNA Differential Display System (GenHunter® Corporation, Nashville, TN, United States). RNA extracted from each MPT embryo (*N* = 3) was then utilized to obtain 1 μg of total RNA, and the same was performed for the FPT embryo (*N* = 3). After extraction, total RNA was reverse-transcribed in two 20 μL reaction mixtures at 37 °C for 60 min using MMLV reverse transcriptase and a set of three one-base anchored oligo(dT) primers (H-T11A/C/G). MPT and FPT cDNA fragments were amplified using combinations of the anchored H-T11 primers from the reverse transcription step and eight different AP upstream primers (Table 2).

The DNA fragments differentially amplified by DDRT-PCR were purified from agarose gel using WIZARDR SV Gel and the PCR Clean-Up System (Promega, Milano, Italy). The purified fragments were T/A inserted into Vector pCRR 4-TOPOR and cloned into *Escherichia coli* DH5α using a TOPOR TA CloningR Kit for Sequencing (Invitrogen, Waltham, MA, USA) according to the manufacturer’s recommendations. Plasmids were extracted by Fast Plasmid Mini Kit (Eppendorf, Hamburg, Germany). Sequencing was performed by Primmbiotech srl (Milan, Italy). Sequences were queried against the NCBI database using Nucleotide BLAST tool and related to known proteins using the tBLASTX algorithm and gene ontology hierarchy [16].

### 2.5. Confirmation of Differential Gene Expression: Real-Time PCR 

qRT-PCR analysis was performed on all the genes of interest, using the same RNA samples employed for the experiments, as previously described. All the primers used for qRT-PCR (Table 1) were designed using the software Primer 3Plus (http://www.bioinformatics.nl/cgiin/primer3plus/primer3plus.cgi, accessed on 10 November 2022). qRT-PCR reactions were carried out using iTaqTM Universal SYBR Green Supermix kit (Bio-Rad, Hercules, CA, USA) in a final reaction volume of 20 μL. For transcript quantification, samples were normalized to the expression level of the endogenous reference gene (GAPDH) to take into account possible differences in cDNA quantity and quality. The amplification protocol involved one cycle at 95 °C for 10 min, to activate Taq DNA polymerase, and 40 cycles consisting of a denaturation step at 95 °C for 15 s and annealing and extension steps at 60 °C for 1 min [16]. Reactions were conducted in an iCycler iQ5 system. The magnitudes of change in gene expression relative to males were determined by the 2^−^^ΔΔ*Ct*^ method of Livak and Schmittgen [18]. Statistical significance was determined using a *t*-test analysis with the Holm–Sidak correction for multiple comparison method using GraphPad Prism 6.0.7 software.

## 3. Results

### 3.1. Genes Involved in Gonadal Sex Differentiation 

Nine transcript fragments of *E. macularius* genes, which are critical for SD in mammals and other vertebrates (*Erα*, *Erβ*, *GnRH*, *P450 aromatase*, *PRL*, *PRL-R*, *WNT4*, *SOX9 and DMRT1*), were amplified by qRT-PCR from RNA of leopard-gecko embryos incubated at sex-specific temperatures and sacrificed at 7 (stage 30) days. In gonads of stage 30, *PRL-R* appeared more expressed in embryos incubated at 26 °C (FPT); *WNT4*, *SOX9* and *DMRT1* were more expressed at 32.5 °C (MPT). *Erα*, *Erβ*, *GnRH* and *P450 aromatase* did not exhibit any statistically significant differential expression (Figure 1). At this stage, we failed to detect expression of *PRL* by qRT-PCR.

### 3.2. Identification and Expression Analysis of Seven New Transcripts by DDRT-PCR 

In analysis of the arbitrary primers provided in the kit, only four (H-AP2, H-AP5, H-AP6, and H-AP7) yielded expression profiles containing bands that were differently expressed (Figure 2). The sequences of these cloned fragments (Appendix A) were compared to those found in Genbank and Embl using BLASTN and TBLASTX. Some correspondence with: *Anolis carolinensis* pyruvate kinase muscle isozyme-like, *Gekko japonicus* aldolase fructose-bisphosphate C (*ALDOC*), *Anolis carolinensis FRAS1-related extracellular matrix protein 1-like*, *Anolis carolinensis BBSome-interacting protein 1-like*, *Sphaerodactylus townsendi* carbonic anhydrase 5A (*CA5A*), *NADH dehydrogenase subunit 5* (mitochondrion) of *Hemitheconyx caudicinctus* and *L1 non-LTR retrotransposons* of *A. carolinensis*, were found (Table 3).

### 3.3. qRT-PCR Expression Analysis of Genes Identified by DDRT-PCR 

All the data collected by DDRT-PCR were validated using qRT-PCR, which partially confirmed the results obtained with the method (Table 4). At stage 30, differential expression was confirmed for six of the seven genes identified: *CA5A* and *L1 non-LTR* expression were stronger in embryos incubated at the FPT, whereas *PKM*, *FREM1*, *BBIP1* and *ALDOC* appeared to be expressed more strongly in MPT. Equal expression levels of *NADH5* were found in male and female embryonic gonads (Figure 3).

## 4. Discussion 

In the animal world, it is usually the genome that provides adequate instructions for the embryonic development of morphological structures; however, in some cases, such as in reptiles, the environment can drive morphogenesis events by modulating the expression of specific genes. Temperature-dependent sex determination (TSD) makes some reptile species ideal models for acquiring information on the space-time path of gene activation. Up to now, the information on the genes that trigger the gene and molecular network at the basis of vertebrate gonadogenesis is still incomplete. Even in mammals, the lack of data on genes expressed during the early stages of gonadal development has limited the possibility of drawing a definitive framework for the genes that would regulate the early stages of ovarian development [12]. It is known that, in the early stages of embryonic development, some important genes involved in sexual differentiation are expressed in a sex-specific way.

*Sox9*, which is part of the family of transcription factors with the HMG box, equivalent to the *SRY* of mammals, is expressed in *E. macularius* only a few days after deposition, at stage 28–30, which, in this species, presumably corresponds to the beginning of the temperature-sensitive period (TSP). *Sox9* could, therefore, have a role in this species in determining the initiation of gonadal differentiation, and its expression found in both gonadal sketches not yet differentiated could therefore trigger this process. Subsequently, its expression decreases in the female embryonic gonad, remaining constant in the male one of some species, until the end of the temperature-sensitive period, or increasing, in others, during the testicular morphogenesis phase [19]. The high degree of identity of the *Sox9* sequence in mammals, birds, fish and reptiles, even in geckos, suggests, however, that there is conservation of its function in all vertebrates.

*Wnt4*, which acts in antagonism with *Sox9*, seems to guide the differentiation pattern of the ovary. Surprisingly, in the analyzed samples, in stages 28 and 29 it was more expressed in the embryo at MDT than in the one held at FDT. In *Trachemis scripta*, the species that gave the greatest amount of information related to the pathway underlying the reptilian TSD, similar expression levels in males and females for *Wnt4* were found at the beginning of the TSP phase, stages 16–19 [20]. In this species, *Wnt4* appears to be over-expressed in females only during ovarian differentiation. 

The expression of *DAX* is also highly variable. In organisms with TSD that have been studied, *Dax* shows a species-specific trend. It is initially expressed at similar levels in MSD and FSD in all species. It was localized, by WISH, both from Muller’s duct and from Wolff’s duct. It then decreases dramatically during embryonic development in *T. scripta* [21] and *L. olivacea* [18]; it increases slightly in the alligator, *A. mississipiensis* [22], and in *C. picta* [23]; but it remains constant in *C. serpentina* [20].

Such a variable trend suggests that Dax has different functions in the various organisms, or that, in these functions, it may have activation timing that does not necessarily correspond in the various species. In the leopard gecko, it was not possible to detect the presence of *DAX* in the early stage we studied; this could agree with an actual involvement of this gene in later stages of gonadal formation. For *Dmrt1*, which is involved in determining the formation of the testicle, the results obtained confirmed what is known in the literature: even if only in minimal quantities, its expression is detectable in males from the beginning of embryonic development [21]. In fact, in situ hybridizations of *Dmrt1* on embryonic gonads of *Podarcis sicula* at 7 days from fertilization show no signals, but the gene is clearly expressed both in the ovary and in the embryonic testis at a later stage, at about 15 days, and then expression is localized only in the testicle until hatching [24]. *Dmrt1* has been the subject of in-depth analysis, as it is considered one of the oldest genes in the sex determination of vertebrates. In fact, genes belonging to the DM family are considered phylogenetically close to *dS* of *D. melanogaster* and *Mab3* of *C. elegans*. The expression of *Dmrt1* is considered not only important for guiding the correct development of the testicle in the embryo but also for maintaining correct testicular function in the adult. Additionally, the expression of the estrogen receptors *Er-b* is clearly present in *E. macularius* embryos from stage 28–30, in both MSD and FSD, without variations. Since these genes are also involved in determining the proper development of the central nervous system and in morphogenesis in general, their early activation may be required to perform these additional functions. During the LP phase (stage 30), the temperature seems to affect aromatase activity and synthesis of estrogens. These data were found not only for the gonads but also for other body structures, such as the brain [17]. Our work is in accordance with previous studies that by utilizing differential display, highlighted how temperatures induced differential expression of several genes involved not only in gonadal differentiation but also, for example, in neural differentiation, in basal metabolic processes or in cell proliferation and differentiation [16]. In our case, we found a group of genes not strictly related to sexual differentiation that also displayed differential expression. One of the differentially expressed sequences, FcAP7, aligned with a portion belonging to the 3′ UTR region of a transposable element present in the genome of a reptile. The data, also validated by analysis for real-time PCR, are interesting because they confirm the presence of repeated elements transcribed during moments of cell differentiation, once again suggesting their probable role in the functionality of the genome. For the Ma1AP5 sequence, on the other hand, we found 83% similarity to the *Anolis carolinensis* pyruvate kinase muscle isozyme-like (*PKM*) sequence, and analysis using the EMBL database revealed similarity with ovarian and testis cDNA libraries of *Anolis carolinensis*. *PKM* is a glycolytic isozyme that catalyzes the transfer of a phosphoryl group from phosphoenolpyruvate to ADP, generating ATP [25]. Different enzymatic forms of it are known [26]; considering their implications in the phenomena of cell growth and proliferation and their involvement in some tumor pathologies [27], it could be hypothesized that the isolated enzyme form is an isozyme expressed only in the phase of embryonic development that could have a role in the induction of cell growth. The *FRAS1-related extracellular matrix 1 transcript*, with which the Ma2AP5 sequence aligns with 95% identity, is associated with craniofacial and renal embryonic formation and development, and its mutation leads precisely to renal agenesis in mice [28]. Given that the gonad and the kidney have the same embryonic origin, one could hypothesize a role of this transcript in the differentiation of the renal portion with respect to the induction of the differentiation of the primordial gonad. The sequence of the transcript of *Anolis carolinensis BBSome-interacting protein 1-like* was instead found following interrogation in the EMBL database (tBLASTx) with the McAP5 sequence (83% identity). We found that the query sequence shows high similarity to *Anolis carolinensis* cDNA libraries derived from transcripts present in the testes and primordial kidneys. The BBSome complex, which contains several isoforms of the BBSome-interacting protein, also forms a protein complex involved in cell trafficking, ciliogenesis and microtubular stability [29]. In Zhang et al. [30], the mutation of this protein is responsible for male infertility in mice due to defects in the formation of the sperm flagellum. Having observed, in the present study, greater expression of the transcript in the male embryo than the female embryo, one could hypothesize a role of *BBSome-interacting protein 1-like* in the induction of male differentiation of the primordial gonad and a default role in both embryos for cell communication. 

Aldolase C fructose-bisphosphate (ALDOC, or ALDC) is an enzyme that, in humans, is encoded by the *ALDOC* gene on chromosome 17. This gene encodes a member of the class I fructose-bisphosphate aldolase gene family [31]. For the first time, this gene was found differentially expressed in male and female embryo gonads. This suggests that there are genes not yet studied during sexual development. *CA5A* (carbonic anhydrase 5A) is a protein-coding gene. Diseases associated with CA5A include carbonic anhydrase Va deficiency, hyperammonemia and carbonic anhydrase Va deficiency. Among its related pathways are metabolism and reversible hydration of carbon dioxide. *CA5A* was shown to be expressed in the ovaries of the Pelibuey breed of sheep; the gene was upregulated in a subset of ewes that gave birth to two lambs compared to uniparous animals [32]. The level of expression led the authors to conclude that *CA5A* is heritable and potentially an imprinted gene [33]. That result is in agreement with our findings. In fact, *CA5A* was more expressed in female embryo gonads.

## 5. Conclusions

Through the present study, new data have been acquired on genes expressed in the early stages of development and sexual differentiation in *E. macularius*. We demonstrated that not only genes related to sexual differentiation, but also genes involved in different developmental pathways, modify their expression in relation to breeding temperature. Our data perfectly fit within the new studies conducted in developmental biology, which indicate that in the developing embryo, in addition to gonadal differentiation, sex-specific tissue and metabolic polarization take place in all organisms. Further investigations will be necessary on embryos at later stages of embryonic development, in order to test the roles of traced transcripts in the determination of gonads and tissues, define any progressive variations in their levels of expression, identify other genes differentially expressed in the later stages of development and analyze their behavior during the reproductive life of the organism.

## Figures and Tables

**Figure 1 animals-12-03186-f001:**
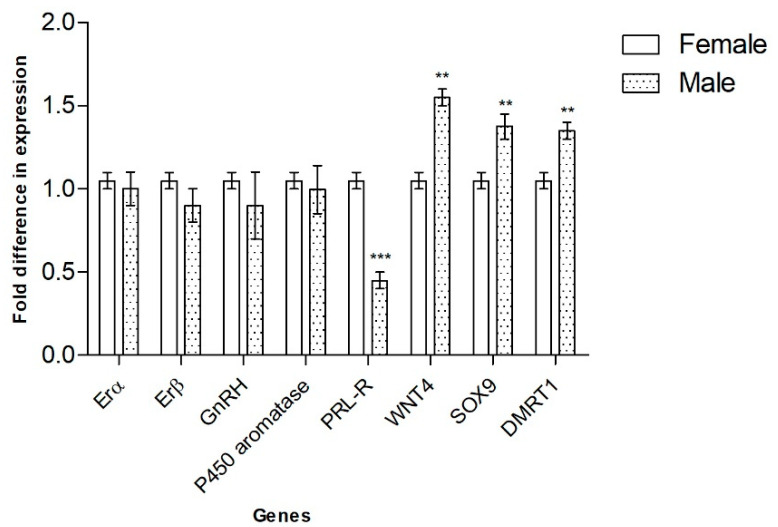
Expression analysis of genes canonically involved in sexual differentiation by qRT-PCR. mRNA levels of FPT embryos are related to MPT embryos. Data are presented as mean with SD. Statistical significance was determined using *t*-tests with Holm–Sidak correction for multiple comparison. ** *p* < 0.01, *** *p* < 0.001.

**Figure 2 animals-12-03186-f002:**
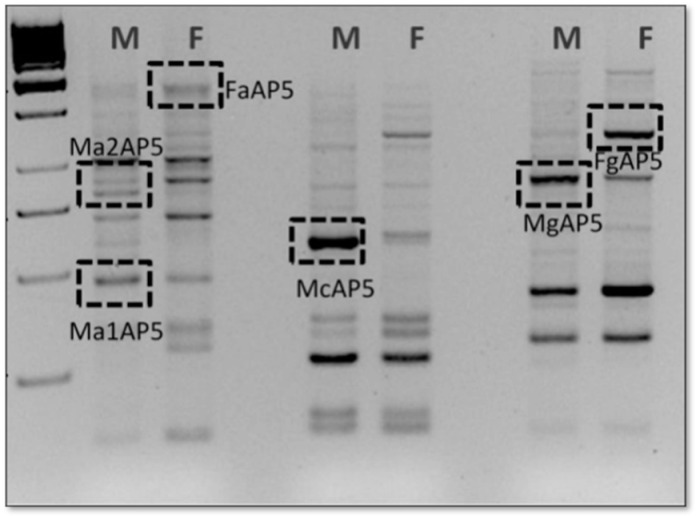
Representative gel image of DDRT-PCR band pattern. The image shows amplifications of cDNA from gonad embryos performed using a 5′ arbitrary primer (H-AP5) in combination with 3′ oligo (dT) H-T11A primer on male (Ma) and female (Fa) embryos.

**Figure 3 animals-12-03186-f003:**
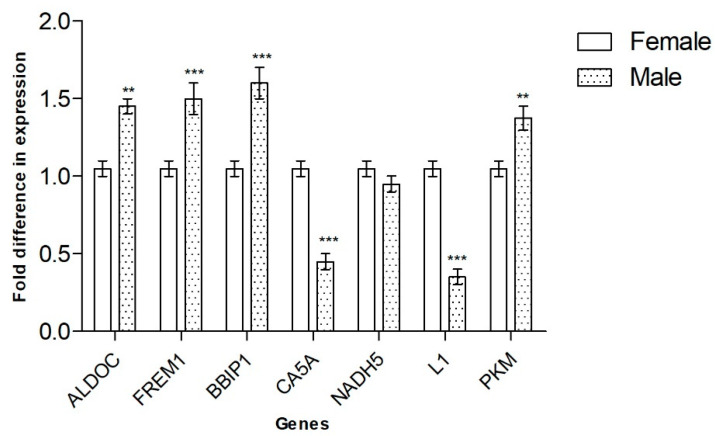
Validation of DDRT-PCR by qRT-PCR. Data are presented as mean with SD. Statistical significance was determined using *t*-tests with Holm–Sidak correction for multiple comparisons. ** *p* < 0.01, *** *p* < 0.001.

**Table 1 animals-12-03186-t001:** Primers used in the qRT-PCR analysis performed to validate the expression profiles of the studied genes.

Gene	Oligo Forward Sequence (5′-3′)	Oligo Reverse Sequence (5′-3′)
*Erα*	CACCCTGGAAAGCTGTTGTT	TTCGGAATCGAGTAGCAGTG
*Erβ*	ATCCCGGCAAGCTAATCTTT	CAGCTCTCGAAACCTTGAAGT
*GnRH*	GTCTTGCTGGCCTCTCCTC	GTGGTCTCCTGCCAGTGTTC
*P450 aromatase*	TGAACACCCTCAGTGTGGAA	TCAGVTTTGGCATGTCTTCA
*PRL*	AAGGCCATGGAGATTGAGG	GGAGGCCTGACCAAGTAGAA
*PRL-R*	ATGGAGGTCTCCCCACTAAT	AACAGGAATTGGGTCCTCCT
*WNT4*	CTGCAACAAGACCTCCAAGG	AGCAGCACCGTGGAATTG
*SOX9*	GGGCAAGCTTTGGAGGTTAC	TGGGCTGGTACTTATAGTCTGGA
*DMRT1*	GCAGGGGATCCTACCAAAGT	AGAAGGCAGCAAGCTCAAGA
*ALDOC*	TACCATGGTGTTGTGCAAGC	CTTCACGCTGCATTTTCTCA
*FREM1*	GGAATGTCAACCAAGATGTGG	CAGGGGAGATCAGAACCACT
*BBIP1*	CGTGAGCTGTAGCTTTGCAG	CTGCCTTACCCACAGCACTT
*CA5A*	TTGCAAAGTTATGGGGAGGA	TCAAGCAGGGTTTATTTCTCATC
*NADH5*	GCTACAGGTAAATCCGCTCAA	AGTAGGGCAGAAACGGGAGT
*L1 non-LTR retrotransposons*	ATCATCGTGGGCCTCTTTGC	AGCAGCACCGTGGAATTG
*PKM*	AGAGCTGCTTGTACGCCTGT	CCAGATTTCCAAAGGACAGTG

**Table 2 animals-12-03186-t002:** Sequences of the 3 oligo (dT) primers and 5 arbitrary primers (H-AP) used in differential display.

Primers	Sequence
3′ oligo (dT) H-T11G	AAGCTTTTTTTTTTTG
3′ oligo (dT) H-T11A	AAGCTTTTTTTTTTTA
3′ oligo (dT) H-T11C	AAGCTTTTTTTTTTTC
H-AP1	AAGCTTGATTGCC
H-AP2	AAGCTTCGACTGT
H-AP3	AAGCTTTGGTCAG
H-AP4	AAGCTTCTCAACG
H-AP5	AAGCTTAGTAGGC
H-AP6	AAGCTTGCACCAT
H-AP7	AAGCTTAACGAGG
H-AP8	AAGCTTTTACCGC

**Table 3 animals-12-03186-t003:** Sequence analysis of differentially expressed mRNAs isolated by DDRT-PCR.

Clone	E-Value	Length (BP)	Identity (%)	Results
McAP2	e−121	450	98%	*Gekko japonicus, aldolase fructose-bisphosphate C (ALDOC)*
Ma1AP5	2e−41	528	83%	*Anolis carolinensis, pyruvate kinase muscle isozyme-like (PKM)*
Ma2AP5	2e−60	710	95%	*Anolis carolinensis, FRAS1 related extracellular matrix 1 (FREM1)*
FaAP5	4e−148	1011	73%	*Hemitheconyx caudicinctus, NADH dehydrogenase subunit 5 (mitochondrion) (NADH-CoQ reduttasii)*
McAP5	1e−144	592	83%	*Anolis carolinensis, BBSome-interacting protein 1-like (BBIP1)*
Fg1AP6	9e−15	137	77%	*Sphaerodactylus townsendi, carbonic anhydrase 5A (CA5A)*
FcAP7	1e−14	202	76%	*Aanolis carolinensis, L1 non-LTR retrotransposons*

**Table 4 animals-12-03186-t004:** Summary of the differentially expressed genes in *E. macularius* embryo at stage 30, listed according to the two different methodologies used. MPT stands for genes more expressed at a male-producing temperature and FPT stands for genes more expressed at a female-producing temperature.

Gene	Identification	Embryo Gonads Stage 30
*Erα*	qRT-PCR	No differential expression
*Erβ*	qRT-PCR	No differential expression
*GnRH*	qRT-PCR	No differential expression
*P450 aromatase*	qRT-PCR	No differential expression
*PRL*	qRT-PCR	Not detected
*PRL-R*	qRT-PCR	FPT
*WNT4*	qRT-PCR	MPT
*SOX9*	qRT-PCR	MPT
*DMRT1*	qRT-PCR	MPT
*ALDOC*	DDRT-PCR	MPT
*FREM1*	DDRT-PCR	MPT
*BBIP1*	DDRT-PCR	MPT
*CA5A*	DDRT-PCR	FPT
*NADH5*	DDRT-PCR	No differential expression
*L1 non-LTR retrotransposons*	DDRT-PCR	FPT
*PKM*	DDRT-PCR	MPT

## Data Availability

The newly generated sequences are available in the Appendix A.

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
