# Peer review of "Temperature Incubation Influences Gonadal Gene Expression during Leopard Gecko Development"

_animals, 2022, doi:10.3390/ani12223186_

Round 1

Reviewer 1 Report

The authors used RT-PCR to determine the expression of genes involved in gonadal sex determination. Besides this, they carried out a differential display RT-PCR. They used q RT-PCR to validate results for both protocols, using trunks of embryos of   E. macularius, incubated at FPT and MPT, stage 30.

The main weakness observed by this reviewer is that although the development of the manuscript focuses on gonadal gene expression, gene expression was carried out for a single stage, in the trunk of gecko embryos. This means that the results do not define the differences observed at the gene expression level of a particular organ, causing them to lose relevance in the context of reported differences.

The only way to corroborate the differences in expression would be to evaluate the differences in expression, by means of in situ hybridization of sections or immunohistochemistry of cross-sections from embryo trunks.

Likewise, the authors should clarify the number of samples used and the number of repetitions implemented.

At the beginning of the simple summary and abstract, topics that are not developed in the main part of the manuscript are addressed.

Author Response

REVIEWER 1: The authors used RT-PCR to determine the expression of genes involved in gonadal sex determination. Besides this, they carried out a differential display RT-PCR. They used q RT-PCR to validate results for both protocols, using trunks of embryos of   E. macularius, incubated at FPT and MPT, stage 30.

The main weakness observed by this reviewer is that although the development of the manuscript focuses on gonadal gene expression, gene expression was carried out for a single stage, in the trunk of gecko embryos. This means that the results do not define the differences observed at the gene expression level of a particular organ, causing them to lose relevance in the context of reported differences.

The embryos were eviscerated and the area strictly adjacent to the not fully formed gonads was taken.  Anyway, research is inherent in sexual differentiation not only of the gonads but of other body structures, as specified in the abstract. As regards the stage, we selected it because it was that of our interest, corresponding to an early stage of development in which the gonad was not completely differentiated (Wise et al.2009). The genes were identified as differentially expressed at stage 30 during the labile phase of sex differentiation (Georges et al. 2010).

The only way to corroborate the differences in expression would be to evaluate the differences in expression, by means of in situ hybridization of sections or immunohistochemistry of cross-sections from embryo trunks.

This suggestion is really interesting. Unfortunately, it is not possible to perform an in-situ hybridization or immunohistochemistry because the number of genes analyzed is very high. In addition, the eggs are available from May to July and therefore, at this time of the year, we do not have material available to repeat the experiment. Despite this, our DDrt-PCR validations are in agreement with other experiments found in the literature. In particular, in these works real-time PCR was used for validating the data (Qu et al. 2015; Pallotta et al. 2017).

Likewise, the authors should clarify the number of samples used and the number of repetitions implemented.

As requested, in Material and Method we rephrased the experiment setting in order to clarify the number of samples and repetition.

At the beginning of the simple summary and abstract, topics that are not developed in the main part of the manuscript are addressed.

Thank you for the suggestion. We modified the abstract and the introduction as requested.

References

Georges A, Ezaz T, Quinn AE, Sarre SD. 2010. Are reptiles predisposed to temperature-dependent sex determination? Sex Dev 4:7–15.

Pallotta MM, Turano M, Ronca R, Mezzasalma M, Petraccioli A, Odierna G, Capriglione T. Brain Gene Expression is Influenced by Incubation Temperature During Leopard Gecko (Eublepharis macularius) Development. J Exp Zool B Mol Dev Evol. 2017 Jun;328(4):360-370.

Wise P, Vickaryous MK, Russell AP. 2009. An embryonic staging table for in ovo development of Eublepharis macularius, the leopard gecko. Anat Rec 292:1198–1212.

Qu XC, Jiang JY, Cheng C, Feng L, Liu QG. Cloning and transcriptional expression of a novel gene during sex inversion of the rice field eel (Monopterus albus). Springerplus. 2015 Dec 1;4:745.

Reviewer 2 Report

This is a very interesting study where authors performed an analysis on E. macular to gain insights into the genes that may be involved during the therm-sensitive period related to its sexual differentiation. This study brings new useful information in order to correctly interpret the regulatory pathway underlying the determination of the sex in vertebrates. Results indicate that in the developing embryo, in addition to gonadal differentiation, sex-specific tissue and metabolic polarization take place in all organisms. In general, the manuscript is absolutely well written and designed. Only minor considerations are listed bellow:

Tittle - Adequate

Summary - Ok

Abstract - Even if it is well written in terms of the presentation of the problematic and justifications for the study execution, it lacks some technical information related the conduction of the study. Please include some information related to the number of individuals, the methodologies used for the exploration, the main objective results found.

Keywords - Please avoid to include indexing terms previously used in the tittle or even into the abstract. Try to use more generic terms as sex determination, incubation temperature, gonadal differentiation, reptile, etc. 

Introduction - This section clearly present justifications for the study execution. After aims, authors included a summary of the methods used. This part should be included in the abstract, not in here.

Methods - Authors are advised to verify the guide for authors and reorganize the tables according to the journal rules. Also, authors should include references for the methodologies used for RNA extraction, gene expression, PCR, etc.

Results - These are well presented. The first column of the Table 4 needs to be revised because the line numbers seems to be covering the data.

Discussion - Authors discussed gene by gene, this is good. Conclusions are adequate.

References - updated an ok.

Author Response

REVIEWER 2: This is a very interesting study where authors performed an analysis on E. macular to gain insights into the genes that may be involved during the therm-sensitive period related to its sexual differentiation. This study brings new useful information in order to correctly interpret the regulatory pathway underlying the determination of the sex in vertebrates. Results indicate that in the developing embryo, in addition to gonadal differentiation, sex-specific tissue and metabolic polarization take place in all organisms. In general, the manuscript is absolutely well written and designed. Only minor considerations are listed bellow:

Tittle - Adequate

Summary - Ok

Abstract - Even if it is well written in terms of the presentation of the problematic and justifications for the study execution, it lacks some technical information related the conduction of the study. Please include some information related to the number of individuals, the methodologies used for the exploration, the main objective results found.

Thank you for the suggestion. As requested, the abstract was revised.

Keywords - Please avoid to include indexing terms previously used in the tittle or even into the abstract. Try to use more generic terms as sex determination, incubation temperature, gonadal differentiation, reptile, etc.

Thank you for the suggestion. Keywords were revised.

Introduction - This section clearly present justifications for the study execution. After aims, authors included a summary of the methods used. This part should be included in the abstract, not in here.

Thank you for the suggestion. As requested, we revised it.

Methods - Authors are advised to verify the guide for authors and reorganize the tables according to the journal rules. Also, authors should include references for the methodologies used for RNA extraction, gene expression, PCR, etc.

Thank you for the suggestion. Tables were reformatted according to journal rules.  References of methods were added.

Results - These are well presented. The first column of the Table 4 needs to be revised because the line numbers seems to be covering the data.

As requested, we revised the tables and therefore the layout of the paper.

Discussion - Authors discussed gene by gene, this is good. Conclusions are adequate.

References - updated an ok.

Round 2

Reviewer 1 Report

Author´s reply:

The embryos were eviscerated and the area strictly adjacent to the not fully formed gonads was taken. Anyway, research is inherent in sexual differentiation not only of the gonads but of other body structures, as specified in the abstract. As regards the stage, we selected it because it was that of our interest, corresponding to an early stage of development in which the gonad was not completely differentiated (Wise et al.2009). The genes were identified as differentially expressed at stage 30 during the labile phase of sex differentiation (Georges et al. 2010).

R: As sexual difference parameters (morphological or histological) in the other “body structures” are not shown (assuming that the authors refer to the urogenital complexes: mesonephros, adrenal gland, gonad, and the remaining neural tissue of the dorsal region of the eviscerated embryos), it is difficult to infer that differences in gene expression detected by DDrt-PCR relate to an effect of temperature on sex. In several biological systems, the temperature can modify the expression of multiple cascades of genes without having a cause-effect relationship on sexual differentiation, for example, concerning metabolic as well as proliferation factors, which indicates the need for the authors to clarify and address this aspect extensively in the discussion.

Author´s reply:

As requested, in Material and Method we rephrased the experiment setting in order to clarify the number of samples and repetition.

R: This section has not yet been clarified; the authors need to specify the following:

N= number of pools. Moreover, defined number of embryos used per pool, both in qRT-PCR and in DDrt-PCR. 

Author Response

As sexual difference parameters (morphological or histological) in the other “body structures” are not shown (assuming that the authors refer to the urogenital complexes: mesonephros, adrenal gland, gonad, and the remaining neural tissue of the dorsal region of the eviscerated embryos), it is difficult to infer that differences in gene expression detected by DDrt-PCR relate to an effect of temperature on sex. In several biological systems, the temperature can modify the expression of multiple cascades of genes without having a cause-effect relationship on sexual differentiation, for example, concerning metabolic as well as proliferation factors, which indicates the need for the authors to clarify and address this aspect extensively in the discussion.

 R: We agree with the referee, it is difficult to associate the variations of all gene expressed as a cause of different sexual differentiation. In this work, we reported a pattern of variation of genes whose expression is generally considered to be sex dependent and of other genes closely associated with them (eg: aromatase). We are aware that data concerning the expression of genes not only linked to sexual differentiation, as happened in our case, can also emerge from the Differential Display. That is why ours is nothing more than a picture of the situation that emerged: the genes linked to differentiation in the male and female sense are certainly varied, the other genes not strictly linked to gonadal differentiation show equally a differential expression. This does not imply that they too are the cause of a difference in the masculine or feminine sense.

As requested we have modified and clarified this point in introduction, discussion and conclusion.

This section has not yet been clarified; the authors need to specify the following:

N= number of pools. Moreover, defined number of embryos used per pool, both in qRT-PCR and in DDrt-PCR. 

R: As requested, it was further specified for the other techniques as well.

Round 3

Reviewer 1 Report

Accept in present form